# Model-Based Reinforcement Learning via Latent-Space Collocation

## Abstract

The ability to plan into the future while utilizing only raw high-dimensional observations, such as images, can provide autonomous agents with broad and general capabilities. However, realistic tasks may require handling sparse rewards and performing temporally extended reasoning, and cannot be solved with only myopic, short-sighted planning. Recent work in model-based reinforcement learning (RL) has shown impressive results using heavily shaped reward functions that require only short-horizon reasoning. In this work, we study how techniques trajectory optimization can enable more effective long-horizon reasoning. We draw on the idea of collocation-based planning and adapt it to the image-based setting by leveraging probabilistic latent variable models, resulting in an algorithm that optimizes trajectories over latent variables. Our latent collocation method (LatCo) provides a general and effective approach to longer-horizon visual planning. Empirically, our approach significantly outperforms prior model-based approaches on challenging visual control tasks with sparse rewards and long-term goals.

## 1 Introduction

In order for autonomous agents to perform complex tasks in open-world settings, they must be able to process high-dimensional sensory inputs, such as images, and reason over long horizons about the potential effects of their actions. Recent work in model-based reinforcement learning (RL) has shown impressive results in autonomous skill acquisition directly from image inputs, demonstrating benefits such as learning efficiency and improved generalization. While these advancements have been largely fueled by improvements on the modeling side – from better uncertainty estimates and incorporation of temporal information (Ebert et al., 2017), to the explicit learning of latent representation spaces (Hafner et al., 2019) – they leave much room for improvement on the planning and optimization side. Most of the current best-performing deep learning approaches for vision-based planning use only gradient-free action sampling as the underlying optimizer, and are typically applied to settings where a dense and well-shaped reward signal is available. In this work, we argue that more powerful planners are necessary for longer-horizon reasoning.

With this goal of long-horizon planning, we aim to extend the myopic planning behavior of existing visual planning methods. Whether it's to avoid local minima due to short-sightedness, or to reason further into the future in order to solve multi-step or sparse-reward tasks, this ability to perform long-horizon planning is critical. Many of the current state-of-the-art visual planning approaches use gradient-free sampling-based optimization methods such as shooting (Ebert et al., 2018; Nagabandi

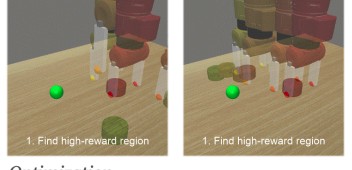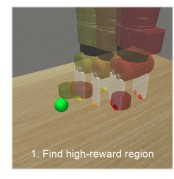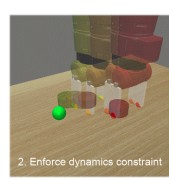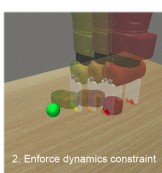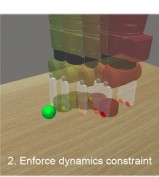

*Optimization*

**Figure 1: Collocation-based planning.** Each image shows a full plan at that optimization step. Collocation jointly optimizes dynamics satisfaction constraints as well as rewards. This ability to violate dynamics allows for the rapid discovery of high-reward regions (where the object is next to the goal), while the subsequent refinement of the planned trajectory focuses on feasibly achieving it.

et al., 2020); as shown in Figure 6, these approaches can get stuck when they must reason further into the future. Here, the curse of dimensionality in conjunction with a lack of shaped reward signal can prevent greedy planning methods from succeeding. Instead, we look to the gradient-based optimization approach of collocation, where crucially, dynamics satisfaction constraints are optimized jointly with the rewards. As shown in Figure 1, this ability to violate dynamics constraints and imagine even impossible trajectories enables collocation to explore much more effectively and take shortcuts through the optimization landscape in order to learn about high-reward regions before figuring out *how* to get to those regions. In contrast to shooting approaches, or even to gradient-based approaches such as backpropagating rewards through model predictions, this ability to violate dynamics greatly helps to prevent getting stuck in local minima while still avoiding the need for special dense or shaped reward signals.

Collocation, as introduced above, can provide many benefits over other optimization techniques, but it has thus far been demonstrated (Ratliff et al., 2009; Mordatch et al., 2012; Schulman et al., 2014) mostly in conjunction with known dynamics models and when performing optimization over states. In this work, we are interested in autonomous skill acquisition directly from image inputs, where both the underlying states as well as the underlying dynamics are unknown. Naïvely applying collocation to this visual planning setting would lead to intractable optimization, due to the high-dimensional as well as partially observed nature of images, in addition to the fact that only a thin manifold in the pixel space constitutes valid images. Instead, we draw from the representation learning literature and leverage latent dynamics models, which learn a latent representation of the observations that is not only Markovian but also compact and lends itself well to planning. In this learned latent space, we propose to perform collocation over states and actions with the joint objective of maximizing rewards as well as minimizing dynamics violations.

Bridging control theory literature with modern deep-RL techniques for learning from images, we propose in this work to perform collocation in a learned latent space in order to enable effective long-horizon planning. The main contribution of this work is an algorithm for latent-space collocation (LatCo), which is an efficient model-based RL approach for solving challenging planning tasks that works directly from image observations by leveraging collocation in a learned latent space. To the best of our knowledge, our paper is the first to scale collocation to visual observations, thus enabling longer-horizon reasoning in model-based RL. In our experiments, we analyze various aspects of our algorithm, and we demonstrate our approach significantly outperforming prior model-based approaches on visual control tasks that require longer-horizon planning with sparse rewards.

## 2 RELATED WORK

**Model-based reinforcement learning.** Recent work has scaled model-based reinforcement learning to complex systems leveraging powerful neural network dynamics models (Chua et al., 2018; Nagabandi et al., 2020), while showing significant data efficiency improvements over model-free agents. Further, these neural network models can be scaled to high-dimensional image observations using convolutional architectures (Ebert et al., 2018; Hafner et al., 2019). However, despite these successes in building better predictive models, planning with these black-box neural network dynamics remains a challenge. While this prior work used simple trajectory optimization techniques like derivative-free shooting, we propose to leverage the more powerful collocation methods. Other work explored more complex approaches based on mixed-integer linear programming (Say et al., 2017) or gradient descent with input-convex neural networks (Chen et al., 2019), but it is unclear whether these approaches scale to visual observations.

**Latent Planning.** Other works have considered different optimization methods such as iterative Linear-Quadratic Regulator (iLQR) (Watter et al., 2015; Zhang et al., 2019). However, these approaches require specialized locally-linear predictive models, and still rely on shooting and local search in the space of actions, which is prone to local minima. Instead, our collocation approach can be used with any latent state model, and is able to optimize in the state-space, which we show often enables us escape local minima and plan better trajectories. Another line of work relied on graph-based optimization (Kurutach et al., 2018; Savinov et al., 2018; Eysenbach et al., 2019; Liu et al., 2020) tree search (Schrittwieser et al., 2019; Parascandolo et al., 2020), or other symbolic planners (Asai & Fukunaga, 2018), while we use continuous optimization, which is more suitable for continuous control. Recent work has designed hierarchical planning methods that plan over extended

periods of time by considering intermediate subgoals (Nair & Finn, 2019; Nasiriany et al., 2019; Pertsch et al., 2020). These approaches are orthogonal to our trajectory optimization method, but they present many possible synergies as collocation can be naturally used in hierarchical approaches. Many of these methods (Buesing et al., 2018; Hafner et al., 2019; 2020) used latent state-space models for improved prediction quality and reduced computational requirements. Our proposed method leverages this latent state-space design to construct an effective trajectory optimization method with collocation, and we design our method to be model-agnostic, such that it can benefit from improved latent variable models in the future.

**Collocation-based planning.** Collocation is a powerful optimal control technique for trajectory optimization (Hargraves & Paris, 1987; Witkin & Kass, 1988) that optimizes a sequence of states for the sum of expected reward, while eventually enforcing the constrained that the optimized trajectory conform to a dynamics model (also see Kelly (2017) for a recent tutorial). Prior work in optimal control has explored many versions of collocation for complex motion planning tasks, including Hamiltonian optimization (Ratliff et al., 2009), contact-invariant optimization (Mordatch et al., 2012), sequential convex programming (Schulman et al., 2014), as well as stochastic trajectory optimization (Kalakrishnan et al., 2011). We note that in this paper we will use the terms "trajectory optimization" and "planning" synonymously referring to this type of approaches. These works have demonstrated good results in controlling complex simulated characters, such as humanoid robots, contact-heavy tasks, and tasks with complex constraints. Our work is most similar to that of Schulman et al. (2014), however, all this prior work assumed availability of a ground truth model of the environment. Some recent works have attempted using collocation with learned neural network dynamics models (Bansal et al., 2016; Du et al., 2019), but it only considered simple low-dimensional dynamics. In this work, we address how to scale up collocation methods to high-dimensional image observations, where direct optimization over images is intractable. We propose to do this by utilizing a learned latent space representation.

## 3 BACKGROUND

Our method combines collocation methods for trajectory optimization with latent variable models, in order to make it possible to optimize paths over compact and Markovian latent states. We separately review collocation methods and latent state models in this section.

### 3.1 TRAJECTORY OPTIMIZATION WITH COLLOCATION

Given a dynamics model $s_{t+1} = f(s_t, a_t)$ that predicts the next state given the previous state and action, a reward function $r(s_t)$, and the current state $s_1$, the problem of trajectory optimization is to select actions that maximize the total reward:

$$\max_{a_{1:T-1}} \sum_t r(s_t) = [r(f(s_1, a_1)) + r(f(f(s_1, a_1), a_2)) + r(f(f(f(s_1, a_1), a_2), a_3) + \ldots].  \quad (1)$$

Shooting methods optimize this objective directly with respect to the actions. However, this is known to be poorly conditioned due to recursive application of the dynamics function, which results in vanishing or exploding gradients. Instead, we can leverage the structure of the problem to construct an objective that only has pairwise dependencies between temporally adjacent states, and no recursive application of the model. To this end, collocation methods formulate the trajectory optimization problem in Equation 1 as a constrained optimization problem, optimizing over sequences of actions and states, while ensuring that the constraint imposed by the dynamics model $f$ is satisfied:

$$\max_{s_{2:T}, a_{1:T-1}} \sum_t r(s_t) \quad \text{s.t.} \quad s_{t+1} = f(s_t, a_t).  \quad (2)$$

This constrained optimization problem, under some regularity conditions, can be solved with a primal dual approach, which can be formulated as the following saddle point problem:

$$\min_\lambda \max_{s_{2:T}, a_{1:T-1}} \sum_t r(s_t) - \lambda ||s_{t+1} - f(s_t, a_t)||^2.  \quad (3)$$

In practice, we can address this problem with numerical optimization, taking alternating maximization and minimization steps. This collocation approach is better conditioned, as it is able to exploit the

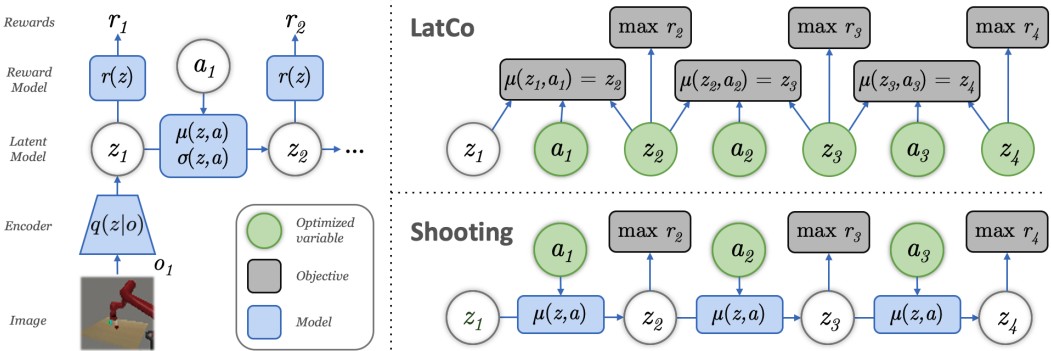

Figure 2: Latent Collocation (LatCo). **Left**: Our latent state-space model, with an encoder $q(z|o)$ and a latent state-space dynamics model $p(z_{t+1}|z_t, a_t) \sim \mathcal{N}(\mu(z_t, a_t), \sigma(z_t, a_t))$. A reward model $r(z_t)$ predicts the reward from the latent state. The model is trained with a variational lower bound to reconstruct the observations (not shown). **Right**: comparison of LatCo and shooting methods. LatCo optimizes a sequence of latent states and actions $z_{2:T}, a_{1:T}$ to maximize rewards $r(z_t)$ as well as satisfy dynamics $z_{t+1} = \mu(z_t, a_t)$. This joint optimization allows to relax the dynamics constraint early on, which helps escape local minima. In contrast, shooting methods require recursive application of the dynamics and backpropagation through time, which is often difficult to optimize.

Markovian property of the state-space and remove the recursive application of the model. The dynamics constraint only needs to be evaluated on each pair of temporally adjacent states (see Fig 2, right). Further, the dynamics constraint can be relaxed in the beginning of the optimization, leading the method to rapidly discover high-reward state space regions (potentially violating the dynamics), and then gradually modifying the trajectory to be more dynamically consistent, as illustrated in Figure 1. This is in contrast to shooting-based approaches, which suffer from severe local optima in long horizon tasks, since the algorithm must simultaneously drive the states toward the high-reward regions *and* discover the actions that will get them there. The ability to disentangle these two stages by first finding the high-reward region and only then optimizing for the actions that achieve that reward allows collocation methods to solve more complex and temporally extended tasks while suffering less from local optima (Ratliff et al., 2009; Mordatch et al., 2012; Schulman et al., 2014).

## 3.2 Latent Variable Models

The design of predictive models for high-dimensional visual observations is challenging. Recent work has proposed learning latent state-space models that represent the observations in a compact latent space. Specifically, this work learns a latent space $z_t$, a latent dynamics model $p_\theta(z_{t+1}|z_t, a_t)$, and decoding models that can be used to decode the observation $p_\theta(o_t|z_t)$ and the reward $r(z_t)$ from the latent variable (see Fig. 2, left). This approach is powerful due to the use of high-capacity neural network latent dynamics models, and is computationally efficient as the latent space is compact. Importantly, the Markov property of the latent variable can be enforced, allowing a convenient interpretation of the latent state-space as the belief state (Hafner et al., 2019; Buesing et al., 2018).

A probabilistic latent variable model can then be trained, by maximizing the likelihood of the observed rewards $r_{1:T}$ and images $o_{1:T}$. While maximizing the exact likelihood is often intractable, we can construct and optimize a variational lower bound on it using a variational distribution $q_\theta(z_{t+1}|o_{t+1}, a_t, z_t)$ (Chung et al., 2015; Fraccaro et al., 2016):

$$\ln p_\theta(o_{2:T}, r_{1:T}|o_1, a_{1:T}) \geq \mathcal{L}_{\text{ELBO}}(o_{1:T}, a_{1:T}, r_{1:T}) =$$

$$\mathbb{E}_{q_\theta(z_{1:T}|o_{1:T}, a_{1:T}, z_0)} \sum_t \left[ \ln p_\theta(o_{t+1}|z_{t+1}) - \text{KL}\left( q_\theta(z_{t+1}|o_{t+1}, a_t, z_t) \,\|\, p_\theta(z_{t+1}|z_t, a_t) \right) \right]. \quad (4)$$

## 4 Latent Collocation (LatCo)

Our aim is to design a collocation method that can be used to produce near-optimal trajectories from raw image observations. A naïve approach would learn a dynamics model over images, and directly optimize an image sequence using Eq (2). However, such as method would be impractical for several

reasons. First, the optimization over images would make the problem more difficult due to the high dimensionality of the images and the fact that valid images lie on a thin manifold. Second, images typically do not constitute a Markovian state space, violating the assumptions of the method. We propose to instead learn a Markovian and compact state space by means of a latent variable model, and then use this learned state space for collocation.

**Latent state models.** To make visual collocation scalable, we leverage the latent state-space dynamics model described in Section 3.2. Following Hafner et al. (2019); Denton & Fergus (2018), we implement this model with convolutional neural networks for the encoder and decoder, and a recurrent neural network for the transition dynamics. The latent state of this model includes a stochastic component with a conditional Gaussian transition function, and the hidden state of the recurrent neural network with deterministic transitions. The model is shown in Fig. 2 (left).

**Latent space collocation.** We adapt the collocation procedure described in Section 3.1 into the probabilistic latent state setting. Instead of optimizing a sequence of observations, we optimize a sequence of latent variables $z_{2:T}$ and actions $a_{1:T}$. Since the latent state space model predicts the reward directly from the latent state $z_t$ using an approximation $r(z_t)$, we never need to decode images during the optimization procedure, which makes it memory-efficient. We adapt the collocation algorithm to use probabilistic dynamics by enforcing that the next latent state is the mean of the distribution predicted by the model: $z_{t+1} = \mathbb{E}_{p_\theta(z_{t+1}|z_t,a_t)}[z_{t+1}]$, where the expectation is given simply as the mean of the Gaussian $z_{t+1} = \mu(z_t, a_t)$. Other approaches are possible, such as maximizing the likelihood of the latent state, but these require introduction of additional balance hyperparameters, while our constrained optimization approach automatically tunes this balance. We visualize the collocation procedure in Fig 2 and provide the detailed algorithm as Algorithm 1.

---

**Algorithm 1** Latent Collocation (LatCo)

---

1: Start with any available data $\mathcal{D}$
2: **while** not converged **do**
3:     **for** each environment step $t = 1 \ldots T_{\text{tot}}$ **with step** $T_{\text{cache}}$ **do**
4:         Infer latent state: $z_t \sim q(z_t|o_t)$
5:         Define the Lagrangian:

$$\mathcal{L}(z_{t+1:t+H}, a_{t:t+H}, \lambda) = \sum_t \left[ r(z_t) - \lambda_t^{\text{dyn}}(||z_{t+1} - \mu(z_t, a_t)||^2 - \epsilon) - \lambda_t^{\text{act}}\max(0, |a_t| - a_m)^2 \right]$$

6:         **for** each optimization step $k = 1 \ldots K$ **do**
7:             Update plan: $z_{t+1:t+H}, a_{t:t+H} := z_{t+1:t+H}, a_{t:t+H} + \tilde{\nabla}\mathcal{L}$       ▷ Eq (5)
8:             Update dual variables: $\lambda_{t:t+H} := \text{UPDATE}(\mathcal{L}, \lambda_{t:t+H})$       ▷ Eq (6)
9:         Execute $a_{t:t+T_{\text{cache}}}$ in environment: $o_{t:t+T_{\text{cache}}}, r_{t:t+T_{\text{cache}}} \sim p_{env}$
10:     Add episode to replay buffer: $\mathcal{D} := \mathcal{D} \cup (o_{1:T_{\text{tot}}}, a_{1:T_{\text{tot}}}, r_{1:T_{\text{tot}}})$
11:     **for** training iteration $i = 1 \ldots \text{It}$ **do**
12:         Sample minibatch from replay buffer: $(o_{1:T}, a_{1:T}, r_{1:T})_{1:b} \sim \mathcal{D}$
13:         Train dynamics model: $\theta := \theta + \alpha\nabla\mathcal{L}_{\text{ELBO}}(o_{1:T}, a_{1:T}, r_{1:T})_{1:b}$       ▷ Eq (4)

---

## 5   Optimization for Latent Collocation

The latent collocation framework described in the previous section can be used with any optimization procedure, such as gradient descent or Adam (Kingma & Ba, 2015). However, using appropriate optimization methods is important when time constraints are a factor. We found that the choice of the optimizer for both the latent states and the Lagrange multipliers has a large influence on runtime performance. We detail our specific implementation below.

**Levenberg-Marquardt optimization.** We use the Levenberg-Marquardt optimizer for the states and actions, which pre-conditions the gradient direction with the matrix $(\mathbf{J}^{\text{T}}\mathbf{J})^{-1}$, where $\mathbf{J}$ is the Jacobian of the objective with respect to the states and actions. This preconditioner approximates the Hessian inverse, significantly improving convergence speed:

$$\tilde{\nabla} = (\mathbf{J}^{\text{T}}\mathbf{J} + \lambda I)^{-1}\mathbf{J}^{\text{T}}\rho. \tag{5}$$

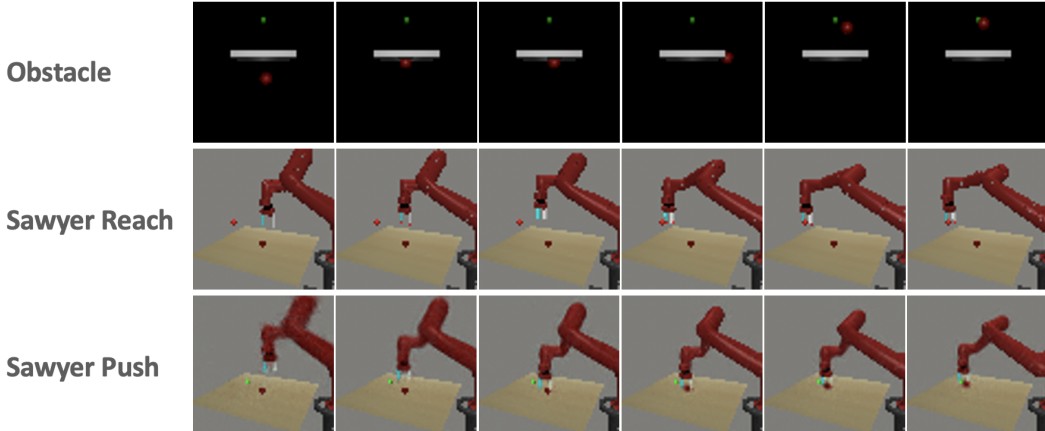

Figure 3: LatCo executed trajectories on the three considered tasks. In all these sparse reward settings requiring temporally extended reasoning, LatCo is able to optimize effective plans and execute them.

The Levenberg-Marquardt optimizer has cubic complexity in the number of optimized dimensions. However, by noting that the Jacobian of the problem has a block-tridiagonal structure, we can implement a more efficient optimizer that scales linearly with the planning horizon (Mordatch et al., 2012). This efficient version of the optimizer converges 10-100 times faster than naïve gradient descent on wall clock time in our experiments.

The Levenberg-Marquardt algorithm optimizes the sum of squares $\sum_i \rho_i^2$, defined in terms of residuals $\rho$. Any bounded objective can be expressed in terms of residuals by a combination shifting and square root operations. For the dynamics objective, we use the $z_{t+1} - \mu(z_t, a_t)$ differences as residuals directly, with one residual per state dimension. For the action objective, we use $\max(0, |a_t| - a_{\mathrm{m}})$. For the reward objective, we form residuals by passing the negative reward through a softplus operation: $\rho = \ln(1 + e^{-r})$.

**Constrained optimization.** The Lagrange multipliers control the balance between the strength of the dynamics constraint and the reward. For fast optimization, it is crucial that the Lagrange multipliers be updated fast when the dynamics constraint is not yet optimized, but the update magnitude should decrease over time to ensure smooth convergence as the dynamic violation converges to zero. We therefore design an update rule that scales with the relative difference between the current dynamics constraint $||z_{t+1} - \mu(z_t, a_t)||^2$ and the target constraint $\epsilon$, such that the update is large when the constraint violation is large and vice versa. We found the following rule to work well in our experiments:

$$\lambda^{\mathrm{dyn}} \mathrel{+}= 0.1 \log \left( \frac{||z_{t+1} - \mu(z_t, a_t)||^2}{\epsilon} + \eta \right) \lambda^{\mathrm{dyn}}, \tag{6}$$

where $\eta = 0.01$ is used to ensure numerical stability. We found that using a small non-zero $\epsilon$ is beneficial for the optimization and ensures fast convergence, as the exact constraint might be hard to reach. We additionally constrain the planned actions to be within the environment range using the same constrained optimization procedure.

## 6 EXPERIMENTS

Our evaluation aims to test the effectiveness of our method on performing long-horizon MBRL from images by answering the following questions: (1) Does LatCo produce effective and feasible plans from visual inputs? (2) Can LatCo solve complex simulated robotic tasks with sparse rewards? (3) How does LatCo compare to trajectory optimization methods used in state-of-the-art visual planning algorithms? (4) What is the effect of LatCo allowing dynamics violations during optimization?

### 6.1 EXPERIMENTAL SETUP

We evaluate our method on three visual tasks (see Fig. 3): *Obstacle*, a navigation environment with an obstacle, *Sawyer Reach*, a robotic reaching task, *Sawyer Push*, a robotic pushing task. The first

of these tasks is a custom environment, while the last two are from the Meta-World benchmark (Yu et al., 2020). All tasks only provide the agent with image inputs and no access to underlying state of the simulator. The Obstacle environment consists of an image-based 2D navigation task where the agent must reach a goal on the other side of a wall. The reward defined is the negative distance to the goal. The agent plans over a planning horizon $H = 30$ steps, executes every action $T_{\text{cache}} = 30, T_{\text{tot}} = 30$, and results are averaged over 100 runs. In the Sawyer Reach task, a 7-DoF Sawyer arm needs to reach the goal. For this task, $H = 30, T_{\text{cache}} = 30, T_{\text{tot}} = 150$. In the Sawyer Push task, a 7-DoF Sawyer arm needs to push an object (puck) on the table to the goal. For this task, $H = 40, T_{\text{cache}} = 20, T_{\text{tot}} = 120$. In both Sawyer tasks, we use a sparse reward of 1 for reaching the goal and 0 otherwise. In particular, for the push task, there is no special reward shaping to encourage the arm to go toward the puck. The arm is initialized to its default position, and results are averaged over 20 runs. We train all methods online according to Algorithm 1. The hyperparameters used in the experiments are detailed in Appendix A.

## 6.2 DOES LATCO PRODUCE EFFECTIVE PLANS ON SPARSE-REWARD VISUAL CONTROL TASKS?

In our first set of experiments, we evaluate how well LatCo can solve image-based control tasks from sparse rewards, where effectively reasoning into the future is critical for success. As shown in Figure 3, LatCo is indeed able to perform collocation in a learned latent space in order to successfully produce plans that solve these tasks, with effective longer-horizon reasoning. In these results, we see LatCo (top) avoiding local minima that would arise from myopic planners, (middle) extending to higher-dimensional control tasks while still operating only from image inputs, and (bottom) reasoning into the future and succeeding even when the sparse task reward provides no direct incentive to go toward the object.

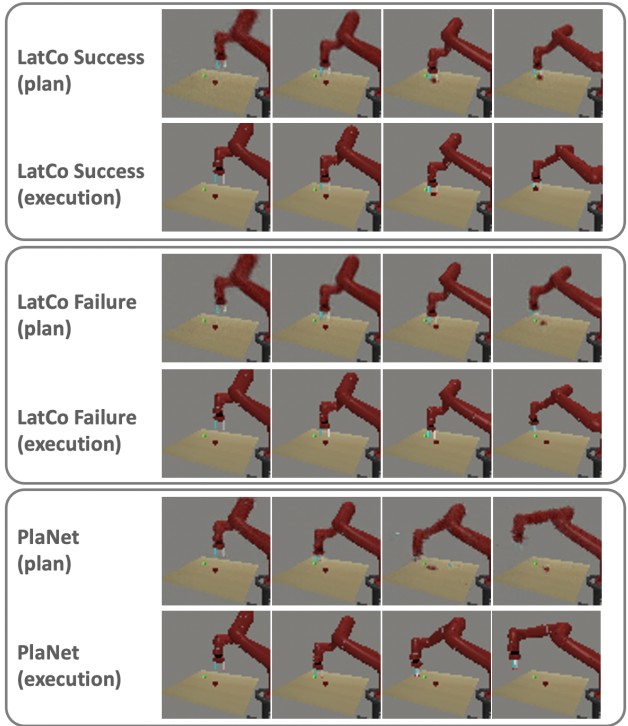

## 6.3 HOW DOES LATCO COMPARE TO ALTERNATIVE SHOOTING-BASED APPROACHES?

To specifically ascertain the benefits of collocation, we must determine whether the benefits of LatCo stem from gradient-based optimization, from optimizing over states, or both. Therefore, we include a prior method based on zeroth-order CEM optimization, PlaNet (Hafner et al., 2019),

Figure 4: Planned and executed trajectories on the Sawyer Push task. LatCo produces a feasible and effective plan, and is able to execute it. Shooting struggles to solve this task as dense or shaped rewards are not provided.

which is representative of a broader class of shooting-based methods (Ebert et al., 2018; Nagabandi et al., 2020), as well as a gradient-based method that optimizes actions directly using the objective in Eq. 1, which we denote *GD actions*. To provide a fair comparison that isolates the effects of different planning methods, we use the same dynamics model architecture for all agents.

From Table 1, we observe that LatCo exhibits superior performance to shooting-based methods on all three tasks. On the Sawyer Reach task, shooting fails to find a successful trajectory most of the time due to a lack of local reward signal. LatCo, on the other hand, solves this task consistently. The Sawyer Push task is challenging for all of the methods, as the dynamics are more complex and requires accurate long-term predictions to correctly optimize the sparse signal. Specifically,

Table 1: Comparison of shooting and collocation with online data. Shooting struggles with long horizon tasks and sparse rewards, while the powerful trajectory optimization with LatCo is able to find a good trajectory.

| | Sawyer Reaching | | Sawyer Pushing | |
|---|---|---|---|---|
| Shaped reward | × | | × | |
| Visual observations | ✓ | | ✓ | |
| | Return | Success | Return | Success |
| Shooting CEM (PlaNet) | 0.0 ±0.0 | 0% | 0.0 ±0.0 | 0% |
| Shooting GD | 38.0 ±5.1 | 42% | 4.3 ±1.5 | 10% |
| LatCo (Ours) | **75.8 ±2.0** | **100%** | **10.9 ±2.1** | **29%** |

Table 2: Comparison of shooting and collocation with offline data.

| | Obstacle | | Sawyer Reaching | | Sawyer Pushing | |
|---|---|---|---|---|---|---|
| Shaped reward | ✓ | | × | | × | |
| Visual observations | ✓ | | ✓ | | ✓ | |
| | Return | Success | Return | Success | Return | Success |
| Shooting CEM (PlaNet) | -52.4 ±0.6 | 25% | 1.4 ±0.8 | 15% | 0.0 ±0.0 | 0% |
| Shooting GD | -54.5 ±0.5 | 34% | 4.8 ±3.8 | 10% | 0.0 ±0.0 | 0% |
| LatCo (Ours) | **-47.3 ±1.1** | **54%** | **90.9 ±5.8** | **100%** | **18.7 ±7.0** | **40%** |

there is no partial reward for reaching down to the object, so the planner has to look ahead into the high reward state and reason backwards that it needs to approach the object. As shown in Fig. 4, shooting-based methods altogether fail to solve this task, not reaching for the object in any run. LatCo outperforms them by a considerable margin, achieving a 29% success rate. It is worth noting that even in the unsuccessful runs, LatCo plans often display goal-reaching behaviors, as can be seen in Fig. 4. However, significant room for improvement for MBRL still remains on this task. In addition, to evaluate the different methods in a more controlled setup, we test them on the same model trained with offline data in Table 2 and App. C

## 6.4 THE EFFECT OF ALLOWING DYNAMICS VIOLATIONS

In this section, we analyze the ability of our method to temporarily violate dynamics in order to effectively plan for long-term reward. We show the dynamics violation costs and the predicted rewards associated with the planned trajectories over the course of optimization quantitatively in Fig. 5 and qualitatively in Figs. 1, 6. Since the dynamics constraint is only gradually enforced with the increase of the Lagrange multipliers, the first few steps of optimization allow for dynamics violations in order to focus on discovering the high-reward region (corresponding to steps 0 to 20 in Fig. 5). In the later steps of the optimization, the reward is relaxed and the constraints are optimized until the trajectory becomes feasible.

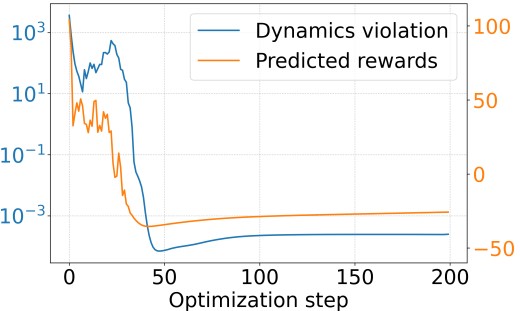

Figure 5: Dynamics violation and reward predictions of planned trajectories over the course of 200 optimization steps on the obstacle task. Earlier on, the planner explores high-reward regions, converging to the goal state. Eventually, as the dynamics constraint is enforced, the plan becomes feasible, while maintaining a high predicted reward.

## 7 CONCLUSION

We presented LatCo, an algorithm for latent collocation that improves performance of visual-model based reinforcement learning agents. In contrast to the commonly used shooting-based methods, LatCo performs powerful trajectory optimization and is able to plan for tasks where prior work fails, such as long-horizon and sparse reward tasks. By improving the planning capabilities of visual model-based reinforcement learning agents and removing the need for reward shaping, LatCo enables these agents to scale to complex tasks more easily and with less manual instrumentation.

Future work will examine improvements to the collocation procedure, such as Hamiltonian (Ratliff et al., 2009) or stochastic (Kalakrishnan et al., 2011) optimization. Further, collocation can be applied with variety of latent variable models, including specialized models with more structure. Finally, we believe the ability to perform reinforcement learning in state space as opposed to action space opens up many new avenues for algorithmic development, such as in imitation learning or hierarchical reinforcement learning.

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

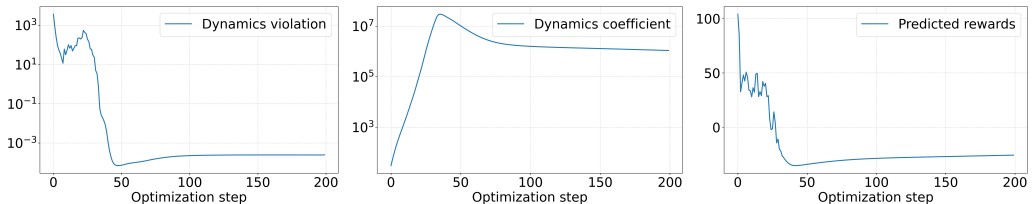

Figure 7: Additional optimization curves. The dynamics coefficient (magnitude of Lagrange multipliers) increases exponentially as the dynamics constraint is enforced, and eventually converges

## A  MODEL ARCHITECTURE AND TRAINING DETAILS

We use the latent dynamics and reward models from PlaNet (Hafner et al., 2019) with default hyperparameters. We set the image size to 64x64 and action repeat to 1 for both pointmass and Meta-World models. For every $N = 1$ episode collected, we train for It $= 15$ iterations. The pointmass models are trained with episode length 30 and $\beta_{KL} = 0.1$, while the Meta-World models are trained with episode length 150 and $\beta_{KL} = 1$. All models are trained on a single high-end GPU.

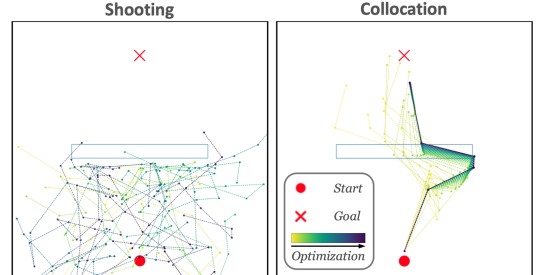

Figure 6: Visualization of collocation-based planning. **Left**: shooting-based planning searches for actions that maximize rewards of the transitions as predicted by a model. It struggles on tasks that require long-term reasoning, as its myopic approach is prone to local minima such as being trapped behind the obstacle here. **Right**: in contrast, collocation jointly optimizes dynamics satisfaction constraints as well as rewards. This ability to violate dynamics allows for the rapid discovery of high-reward regions (here, the goal state), while the subsequent refinement of the planned trajectory focuses on feasibly achieving it.

## B  PLANNING DETAILS

**CEM.**  we optimize for 100 iterations. In each iteration, 10000 action sequences are sampled and the distribution is refit to the 100 best samples. For this baseline, we have manually tuned the batch size and number of iterations, and we report the best results.

**GD.**  we optimize for 500 iterations using the Adam optimizer (Kingma & Ba, 2015) (which is a modified version of momentum gradient descent) with learning rate 0.05. We use dual descent to penalize infeasible action predictions. The Lagrange multipliers are updated every 5 optimization steps. For this baseline, we have manually tuned the learning rate and tried several first-order optimizers, and we report the best results.

**LatCo.**  we optimize for 200 iterations using the Levenberg-Marquardt optimizer with damping $10^{-3}$. The damping parameter controls the trust region, with smaller or zero damping speeding up convergence, but potentially leading to numerical instability or divergence. The Lagrange multipliers are updated every step using the rule from Section 5, with $\epsilon = 10^{-5}$ and $\eta = 0.01$. The threshold $\epsilon$ directly controls the magnitude of the final dynamics and action violations. In general, we found this parameter to be important for good performance, as a large threshold may cause infeasible plans, while low threshold would make the initial relaxation of the dynamics constraint less effective. However, we observed that a single threshold of $10^{-5}$ works for all of our environments. A larger $\eta$ makes the optimization more aggressive but less stable, and a smaller $\eta$ diminishes the effect of multiplier updates. We initilalize $\lambda_0^{dyn} = 1$, $\lambda_0^{act} = 1$.

These planning hyperparameters remain fixed across the experiments as we observe that reward optimization converges in all cases. Planning a 30-step trajectory takes 12, 14, and 14 seconds for CEM, GD, and LatCo respectively on a single RTX 2080Ti graphics card. We trained the online models for 24 hours each.

Table 3: Comparison of predicted and achieved reward.

| | Obstacle | | Sawyer Reaching | | Sawyer Pushing | |
|---|---|---|---|---|---|---|
| | Predicted | Real | Predicted | Real | Predicted | Real |
| Shooting CEM (PlaNet) | -8.3 ±0.1 | -54.7 ±2.3 | 22.1 ±2.3 | 0.0 ±0.0 | 3.2 ±0.0 | 0.0 ±0.0 |
| Shooting GD | -40.8 ±2.3 | -49.3 ±2.0 | 5.9 ±0.7 | 0.0 ±0.0 | 3.0 ±0.0 | 0.0 ±0.0 |
| LatCo (Ours) | -20.7 ±0.9 | **-46.9 ±2.7** | 141.4 ±2.8 | **90.8 ±4.7** | 29.3 ±5.9 | **18.2 ±6.5** |

## C  ADDITIONAL EXPERIMENTAL RESULTS

**Offline training.**    In Section 6.3, we have evaluated our algorithm in the online training regime, where each agent needs to collect the entire dataset from scratch by executing the policy in the environment. To further test whether collocation is beneficial for planning in a controlled setup, we have performed offline training experiments, where a single model trained with offline data is tested with different planning algorithms. This experiment further shows the applicability of our method to offline setting without additional data collection. We collect the offline data by training an oracle agent that observes dense reward, avoiding the exploration issue. We pre-collect 200K steps of this data for all tasks. For the Sawyer Push task, we collect an additional 200K environments steps of online interaction data according to Algorithm 1 as none of the methods were able to solve the task with offline data only. We use the same hyperparameters as in Section 6.3 otherwise.

Figure 8: Visualization of the reward predictor for the Sawyer Pushing task. The output of the reward predictor is shown for each object position on the 2D table. We see that the reward predictor correctly predicts a value of 1 at the goal, and low values otherwise. In addition, there is a certain amount of smoothing induced by the reward predictor, which creates a gradient from the start to the goal position. This explains why gradient-based planning is applicable even in this sparse reward task.

Table 2 shows the results of this experiment. We see that collocation outperforms shooting methods in this controlled setup, showing that it is better able to exploit a dynamics model trained offline.

We visualize the optimization procedure on the obstacle task in Fig 6.

We visualize the reward predictor output on the Pushing task in Fig 8.

We visualize the additional curves of the optimization in Fig 7.

We further compare the predicted and achieved reward by all methods in the offline setup in Table 3. The predicted reward is often higher than the achieved reward, indicating some degree of model exploitation. However, we see that the predicted reward reflects the general trends in the achieved reward and trajectories with high predicted reward also achieve high reward for our method.

