# OpenReview forum: "Model-Based Reinforcement Learning via Latent-Space Collocation"
_ICLR.cc/2021/Conference — Reject_

### Official Review · AnonReviewer4 · 2020-10-24
**Review of LatCo**

**Rating:** 7
**Confidence:** 4

**Review:**

Summary: The paper studies the problem of planning in domains with sparse rewards where observations are in the form of images. It focuses on solving this problem using model-based RL with emphasis on better trajectory optimization. The proposed solution uses latent models to extract latent representations of the planning problem that is optimized using the Levenberg-Marquardt algorithm (over a horizon). The experimental results show improvements over a) zeroth-order CEM optimization, b)  PlaNet (Hafner et al., 2019) and c)  gradient-based method that optimizes the objective in Eq. 1.

Strengths:

i) The motivation, organization and the overall writing of the paper are clear.

ii) The tested experimental domains are good representatives of the realistic planning setting identified in the paper.

Weaknesses:

i) Discussion of literature on planning in latent spaces [1,2,3,4,5] is left out and should be included. Namely, [1,2] performs (classical) planning from images, and [3,4,5] perform planning with learned neural models. Here, space can be saved by removing Figure 4 since all of its subfigures look identical given their (visual) quality.

ii) Have you tried solving Eq. 2. directly similar to [4]? It seems more appropriate baseline compared to c) (i.e., as labeled above).

iii) How do you reason about the length of the horizon T? For example [1,2] use heuristic search.

iv) There does not seem to be any presentation of hyperparameter selection/optimization, runtime results or quality of solutions. Table 1 is too high-level to provide any meaningful insight into understanding how each method compares. Similarly, Figure 5 is very hard to read and not clear what each axis represents. Overall, I would say this is the weakest part of the paper.

References:

[1] Classical Planning in Deep Latent Space: Bridging the Subsymbolic-Symbolic Boundary, Asai and Fukunaga AAAI-18.

[2] Learning Neural-Symbolic Descriptive Planning Models via Cube-Space Priors: The Voyage Home (to STRIPS), Asai and Muise IJCAI-20.

[3] Nonlinear Hybrid Planning with Deep Net Learned Transition Models and Mixed-Integer Linear Programming, Say et al., IJCAI-17.

[4] Scalable Planning with Deep Neural Network Learned Transition Models, Wu et al. JAIR.

[5] Optimal Control Via Neural Networks: A Convex Approach, Chen et al., ICLR 2019.

** Post Rebuttal **

To best of my understanding, the authors have addressed all my questions and suggestions with the appropriate revision of their paper. Specifically, the necessary discussion of hyperparameter selection is added and presentation of the runtime&solution quality results (i.e., raised in point iv)) have been improved with the inclusion of important details, additional discussion of related work is added (i.e., raised in point i)) and questions are addressed (i.e., raised in point ii) and iii)). As such, I have updated my rating accordingly.

---

> ### Author Response · Authors · 2020-11-19
> **Author Response: Improved paper presentation**
>
> We thank the reviewer for the valuable feedback. We believe that the main concern raised in the review is the qualitative evaluation of the compared methods. We have improved the qualitative evaluation in the revised submission, including several new visualizations, as also detailed below. We would like to ask the reviewer whether these revisions adequately address the raised issues.
>
> _There does not seem to be any presentation of hyperparameter selection/optimization, runtime results or quality of solutions._
> The hyperparameters were tuned manually. We added an explanation of hyperparameter selection to Appendix B. Note that we use the same hyperparameters for all environments, which suggests that these hyperparameters are robust. The analysis of runtime is also in Appendix B. The quality of solutions is shown qualitatively in Figs 3,4 as well as on the supplementary website, and quantitatively evaluated in Tab 1. Further, we have produced a higher quality visualization in Fig 1 and on the website. We have also improved Fig 5 and labeled the axes. We hope that these changes resolve the reviewer’s questions and are happy to provide any other clarifications.
>
> _Have you tried solving Eq. 2. directly ... ?_
> All compared methods (a, b, c in the classification of the reviewer) solve Eq 2 directly using different optimization methods and parameterizations. Please see Section 3.1 that summarizes the compared approaches, or Kelly’17 for a recent tutorial on trajectory optimization.
>
> _How do you reason about the length of the horizon T? For example [1,2] use heuristic search._
> As in prior work (e.g. Ebert’18, Hafner’19), horizon T was chosen manually for each problem such that the problem is solvable by extending the horizon until the problem is solved.
>
> _Discussion of literature on planning in latent spaces [1,2,3,4,5] is left out._
> We thank the reviewer for pointing out these interesting papers. We would like to ask the reviewer to clarify the relevance of these papers. As far as we understand, [3,4,5] do not do planning in latent spaces, and are similar to the representative papers by Chua’18, Nagabandi’20 that we cite for non-latent planning. [1,2] address discrete MDP, while we focus on planning in continuous MDPs and review prior work that addresses continuous MDPs.

---

> > ### Comment · AnonReviewer4 · 2020-11-19
> > **Review of LatCo**
> >
> > Thank you for your responses; let me try to address your comments and questions below:
> >
> > Runtime and Solution Presentation: In Appendix B, there are only 3 results presented for runtime (i.e., Planning a 30-step trajectory takes 12, 14, and 14 seconds for CEM, GD, and LatCo respectively). Is it the case that the paper solves only 1 instance per method and time step (i.e., CEM, GD, and LatCo for T=30)? This cannot be right so please explain. Moreover, I do not see a graph or a table (or any other information in the text in Appendix B) that shows the total reward collected by solving each optimization method (i.e., in units of reward). Is "return" used for "total reward"? Regardless, assuming you have not solved 1 instance per method and T=30, these results should have some measure of variance.
> >
> > Discussion of Literature on Planning: Their relevance is that they solve real-valued factored action and state spaces, as you do in this paper, by solving Eq. 2 directly for learned neural state transition models. The main difference is that there is an additional level of state-abstraction in your paper (i.e., in the form of an encoder and decoder). This was a minor point really; just to make the paper relevant to a greater audience.

---

> > > ### Author Response · Authors · 2020-11-19
> > > **Authors Response: further improvements and clarifications**
> > >
> > > We thank the reviewer for the quick response!
> > >
> > > _In Appendix B, there are only 3 results presented for runtime (i.e., Planning a 30-step trajectory takes 12, 14, and 14 seconds for CEM, GD, and LatCo respectively). Is it the case that the paper solves only 1 instance per method and time step (i.e., CEM, GD, and LatCo for T=30)? This cannot be right so please explain._
> > > We report the average runtime of a single optimization problem. The variance of runtime is negligible since it is only dependent on hardware. As per Algorithm 1, our algorithm solves one optimization problem per $T_{rep}$ time steps, where $T_{rep}$ is 20 or 30, depending on the task. A single episode therefore takes 72, 84, 84 seconds respectively to optimize on the Pushing task. We note that this is measured on our working research implementation and we expect that optimization can be significantly sped up. We have now added to the appendix B that we train the online models for 24 hours following Algorithm 1 (i.e., this includes both model training and data collection). Please let us know if this addresses your question and what other information we could provide.
> > >
> > > _Total reward collected by solving each optimization method._
> > > We indeed use ‘return’ to denote total reward, as is common in reinforcement learning. Note that ‘return’ is the result of executing the planned trajectory in the real environment, which is also averaged across 20 runs. The predicted reward of the plan (i.e. the value that is being optimized) is shown in Fig 5 for the obstacle task.
> > >
> > > _Results should have some measure of variance._
> > > The results in the tables are indeed averages across runs. We added the standard error of the mean to the tables.
> > >
> > > _Discussion of Literature on Planning_
> > > Thank you for the detailed explanation! We will review this literature closer so that we can cite the appropriate papers.

---

> > > > ### Comment · AnonReviewer4 · 2020-11-19
> > > > **Review of LatCo**
> > > >
> > > > Thank you for addressing my comments and questions; I think the proposed changes and additions will make the paper stronger.
> > > >
> > > > Looking at the returned values (from the environment), it is surprising (to me at least, and any insight you have here is very valuable) that GD is performing so poorly on even smaller problems (e.g., obstacle domain). It might be the case that GD requires better tuning of hyperparameters (and/or simply need to use another optimizer e.g., RMSProp) as investigated in paper [6] (i.e., see sections 3.2.4 and 3.2.5).
> > > >
> > > > Reference:
> > > >
> > > > [6] Scalable Planning with Tensorflow for Hybrid Nonlinear Domains, Wu et al. NeurIPS 2017.

---

> > > > > ### Author Response · Authors · 2020-11-19
> > > > > **Author response: GD shooting baseline**
> > > > >
> > > > >
> > > > > Thanks, we agree that the suggested changes improve the paper!
> > > > >
> > > > > _GD is performing so poorly on even smaller problems (e.g., obstacle domain)_
> > > > > We note that GD shooting does solve the obstacle task sometimes, as well as the reaching task in the online setting. However, it often fails even these tasks since the shooting methods are prone to getting stuck in local minima (see Fig 5 and qualitative examples on the website for the solutions). We have tuned the learning rate for this baseline and tried several first-order optimization methods such as Adam, and report the best results. We did not observe significant improvements from different optimizers, consistent with Wu'17, which reports that different optimizers improve convergence, while our baselines instead struggle with local minima. We also note that the GD shooting method performs on par or better than the current state-of-the-art CEM shooting method (e.g. used in Ebert’18, Hafner’19, Nagabandi’20). We thus believe that our baselines are representative of currently used methods.

---

> > > > > > ### Comment · AnonReviewer4 · 2020-11-19
> > > > > > **Review of LatCo**
> > > > > >
> > > > > > Thank you for your response.
> > > > > >
> > > > > > Again, it is helpful to note in the paper that you have tried hyperparameter optimization and different optimizers, and have not observed a significant difference in returns for your baselines (and/or reported the results of best combination of hyperparameter&optimizer). It should be clear from reading the paper that a straightforward hyperparameter search does not yield significantly better results - especially given these problems take 10 to 100 seconds to solve.
> > > > > >
> > > > > > Finally, please try to address all discussions from all reviewers in the final version of your paper to best of your ability.

---

> > > > > > > ### Author Response · Authors · 2020-11-20
> > > > > > > **Author Response**
> > > > > > >
> > > > > > > We thank the reviewer for the detailed feedback. We would like to ask the reviewer whether there are any remaining concerns after our revisions, and to update the score if the concerns have been resolved.
> > > > > > >
> > > > > > > _It should be clear from reading the paper that a straightforward hyperparameter search does not yield significantly better results_
> > > > > > > We have now clarified that we tune hyperparameters for all methods in App B.
> > > > > > >
> > > > > > > _Finally, please try to address all discussions from all reviewers in the final version of your paper to best of your ability._
> > > > > > > We have now added the discussion of planning papers suggested by the reviewer. We believe we did address all comments to the best of our ability. Please let us know if we missed any of the suggestions.

---

> > > > > > > > ### Comment · AnonReviewer4 · 2020-11-20
> > > > > > > > **Review of LatCo**
> > > > > > > >
> > > > > > > > Thank you; to best of my understanding you have addressed all my questions and concerns.
> > > > > > > >
> > > > > > > > One final questions/suggestion I have is the following: The results presented in Appendix C is very useful (i.e., especially Table 2). Is it also possible to include information on the predicted total rewards per domain per method using the same format of the return column? This is really a sanity check of your results to rule out the possibility that high return values are due to high predicted total rewards and not due to prediction errors of the underlying learned model.

---

> > > > > > > > > ### Author Response · Authors · 2020-11-22
> > > > > > > > > **Author Response**
> > > > > > > > >
> > > > > > > > > _Include information on the predicted total rewards_
> > > > > > > > > We have included the comparison of predicted and total reward in Tab 3. The predicted reward is often higher than the achieved reward, indicating some degree of model exploitation. However, we see that the predicted reward reflects the general trends in the achieved reward, and trajectories with high predicted rewards also achieve high rewards for our method.
> > > > > > > > >
> > > > > > > > > _"High return values are due to high predicted total rewards" or "errors of the underlying learned model"_
> > > > > > > > > We indeed see that trajectories with high predicted rewards also achieve high rewards for our method.

---

### Official Review · AnonReviewer1 · 2020-10-27
**Sensible idea, some concerns about method clarity**

**Rating:** 6
**Confidence:** 4

**Review:**

## Paper summary

This paper introduces a vision-based motion planning approach using collocation. Many existing approaches to vision-based control rely on computationally expensive planning approaches using shooting to perform model-based control, which is often only useful in simple control tasks. Collocation approaches are effective in settings with difficult path constraints, and thus exploited by this work to dramatically improve model-based reinforcement learning.

I like the idea, but it is a relatively small extension to existing work, so I am inclined to rate this paper as marginally below the acceptance threshold. I would be willing to revise my score if the paper was revised to
- better clarify the algorithm to align with the methods used in experiments
- better justify the reasons why ilQR trajectory optimisation with locally linear dynamics models was not used as a baseline  (or even better, include this as a baseline)

### Pros

- The paper is well written and clearly laid out.
- Solving a collocation problem in the latent space is a sensible approach, and a much better idea than using CEM planning or shooting.

### Cons
- It's a reasonably straightforward application of collocation in a learned latent space. While I have not seen this done previously, it is a relatively obvious improvement.
- The paper motivates the need for collocation in the context of *long horizon tasks*, where shooting performs poorly. However, none of the tasks (pushing and reaching in free space) considered in this work are long horizon tasks, or particularly challenging.

### General recommendations for improvement and queries

-  I'd recommend replacing the term *long horizon tasks* with something more suitable, along the lines of what is actually demonstrated in the experimental results, eg. *vision-based motion planning*.
- Page 2 - Latent Planning.  The paper mentions work on structured latent dynamical systems (Watter  et al. 15), but disregards these "*However, these approaches relied on locally-linear predictive models, which may be difficult to design.*" No design is required for latent dynamical systems with local linear latent dynamics (eg. Watter  et al. 15, [Fraccaro et al. 17](https://arxiv.org/pdf/1710.05741.pdf)) - all transition matrices and parameters are learned, using a slightly different ELBO. The benefit of this approach is that it allows for standard trajectory optimisation approaches like iLQR to be applied directly. I would like to see a comparison against trajectory optimisation using a dynamical system with learned locally linear models, which arguably allows for simpler planning and control.
- Along the lines above, there is a recent body of work looking at imposing more structure in the latent dynamical system to simplify and improve downstream control (eg. embedding for proportionality - [Jaques et al.](https://arxiv.org/abs/2006.01959), koopman embeddings for open loop control with QP [Li et al.](https://openreview.net/forum?id=H1ldzA4tPr) In contrast, this work seems to advocate the opposite approach - ignoring the latent dynamical system learned, and focusing on better methods to solve a more challenging optimisation problem. I  believe that more discussion on the contrasts between these ideas would be a useful addition to this paper.
- Algorithm 1. The algorithm and training approaches lack clarity and cause some confusion, which needs to be improved. The algorithm seems to indicate that dynamics model learning and planning happen jointly,  which doesn't really make sense - we shouldn't need to re-learn a dynamics model at planning time. Unless the intention was to imply that this is an online learning approach? I assume that this is not the case, as experimental methods seem to indicate that dynamics and reward models are pre-trained, separately from trajectory optimisation using collocation. Please clarify, and ensure that the methodology lines up with what was demonstrated in the experiments section.

---

> ### Author Response · Authors · 2020-11-19
> **Author Response: Provided online training experiment, related work on structured models discussion**
>
> We thank the reviewer for the insightful and helpful feedback. The main concerns raised in the review were about online training of the model and differences with iLQR. To address the concerns, we have now performed experiments with online training that don’t require oracle data, and corrected the description of iLQR methods. We explain our revisions below in detail.
>
> _”The intention was to imply that this is an online learning approach?”_
> The algorithm describes our proposed model-based reinforcement learning approach, which is an online agent. This agent is used in the experiments on the Sawyer Pushing task, where online learning is necessary to correct the inaccuracies of the model. To avoid confusion, we have now re-run our experiments with online training only instead of the oracle data that were used previously and updated Tab 1 with these results. On this harder evaluation protocol due to the additional exploration problem, our agent still outperforms shooting-based methods, which are not able to solve the robotic tasks.
>
> _“Better justify the reasons why ... locally linear dynamics models was not used as a baseline”_
> Thanks for catching this! We have corrected the description of these methods. We also stress that these methods still use shooting as they perform local search in the action space. Instead, we use collocation and show that it suffers less from local minima than some shooting-based methods. We were not able to perform a comparison to Watter’15 since the publicly available implementations work poorly and do not replicate the results in the original paper.
>
> _“Focusing on better ... optimisation” instead of better latent models_
> Indeed, we believe our paper fills in a significant gap in the literature, as most papers focus on improvements in the predictive model or structure of the latent space, while we show that performance can be significantly improved by using more appropriate optimization methods for planning. We discuss this in the first paragraph of the introduction. We do not believe that these ideas are in contrast; in fact, we expect our method to benefit from newer latent variable models, and we have added a discussion of this to the related work section and the conclusion.
>
> _”None of the tasks ... are long horizon tasks, or particularly challenging.”_
> While our tasks have indeed similar episode length to the tasks usually considered in RL, we note that they in fact require long-horizon reasoning due to sparse rewards.  Typical benchmark tasks use dense rewards, meaning that the agent does not need to plan ahead, but can often simply maximize the immediate reward greedily and still succeed. In our work, we show that current model-based reinforcement learning methods that use shooting fail on sparse reward tasks, as they are not able to plan that long ahead (15-30 steps until reward is seen). By designing a method that is able to plan on these sparse reward tasks, we believe our work constitutes an important step toward tackling more challenging and long-horizon tasks.

---

> > ### Comment · AnonReviewer1 · 2020-11-19
> > **Thank you for your response**
> >
> > Thank you for responding to my review. Thank you too for re-running the experiments to align with Algorithm 1. However, I note that the values in the table have changed greatly. Specifically,
> > - Why has the return changed so much?
> > - Why was the obstacle task removed from Table 1?

---

> > > ### Author Response · Authors · 2020-11-19
> > > **Author Response**
> > >
> > > Thanks for the quick response!
> > >
> > > _Why has the return changed so much?_
> > > We have switched from reporting the dense reward (also defined by Meta-World) to sparse reward for consistency. Please compare the online training experiment in Tab 1 to the offline experiment now in Tab 2. The performance with offline data is slightly better since offline data help avoid the exploration challenge. We now include both experiments, with the online experiment highlighting the application to model-based RL, and the offline experiment serving as a controlled setup to disentangle the planning and exploration.
> > >
> > > _Why was the obstacle task removed from Table 1?_
> > > The previous obstacle task experiment is still present in Tab 2. We are working on an online experiment on the obstacle task and will add it in the next revision.

---

### Official Review · AnonReviewer2 · 2020-10-28
**useful idea;  uninteresting examples**

**Rating:** 6
**Confidence:** 4

**Review:**

## summary
The paper proposes to transpose colloction methods to solve planning problems in a learned latent state space.
This can then be used as a replacement for shooting methods in model-based RL, particularly suitable
for image-based tasks, where planning in the observation space is impractical.

## pros
- Basic shooting methods are a primitive planning technique; we should be able to do much better.
  Using collocation methods in learned latent state spaces makes sense. This paper is one of the first
  to provide a working realization of this.

## cons
- The problem is only difficult because of the attempt to learn the task directly from visual inputs.
  From a practical robotics and planning perspective, the task problems are very dated, e.g., from 30 years ago.
  In this sense, the tasks are "straw man" problems that are uninspiring.
- Shooting methods provide exploration that the gradient-driven collocation methods do not allow for.
  The tradeoffs are not as simple as portrayed.

## recommendations
I currently lean marginally in favor of acceptance, purely on the grounds that transposing collocation
methods to latent spaces does havae future potential.  However, the given examples are uninteresting.

## questions
- How would the results compare to simply using the latent state to estimate a traditional compact state descriptor
  and then using that with a classical motion planner? For the given example tasks, that seems very feasible.
- Can planning methods like CHOMP also be realized in the latent space?
  What are the general constraints or restrictions, if any, on transposing the many known planning methods into
  the latent space?
- What is the impact of choosing a time horizon T that is too short or too long?
- What is stochastic about the dynamics, if anything, for the chosen experimental tasks?
- What is the action space for the given tasks? What is a-max for the tasks?


## feedback
The output is a trajectory, not a policy. To make it actionable would require using the optimized
trajectories to learn a policy or to use MPC.  This aspect is missing from the paper. Similarly,
the exploration issue is avoided (cf sec 6.1).  Thus, overall, the paper is not really solving an
RL problem.  The title could more directly address the contribution, i.e., motion planning via
latent-space collocation.

"To this, collocation methods" (sic)

Figure 2: the text refers to a decoder, but this is missing in the figure.
The dynamics model is left unlabeled.

It is worthwhile briefly discussing the broader space of collocation methods, and where your method
fits within that taxonomy.

Section 5, Constrained optimization: "balance between the strength of the dynamics constraint."
missing: "and the objective" ?

---

> ### Author Response · Authors · 2020-11-19
> **Author Response: Provided online training experiment, related work on collocation discussion**
>
> We thank the reviewer for the valuable feedback and suggestions, which we address individually below.
>
> _”The paper is not really solving an RL problem”_
> We selected the original evaluation protocol to isolate the effect of planning from exploration and model learning. To address the reviewer’s concern, we have now added an experiment that trains the agents according to Algorithm 1 (using MPC and online training), without any offline data. We have updated Tab 1 with these results. On this harder evaluation protocol due to the additional exploration problem, our agent still outperforms shooting-based methods, which are not able to solve the robotic tasks.
>
> _“From a practical robotics and planning perspective, the task problems are very dated”_
> We agree with the general sentiment that using ground truth state information and dynamics models would make many current RL tasks uninteresting. However, we would like to point out that we evaluate on standard tasks in current RL research, such as navigation, robotic control and robotic manipulation. The tasks we use were proposed at the conference for robot learning last year (Yu’2019). Indeed, we note that these sparse reward tasks are too challenging for current visual planning methods (e.g. Hafner’19), which completely fail some of the tasks. Our aim in choosing these tasks is to evaluate the hypotheses put forward in the paper, which focus on planning with learned models and with image observations. We did not intend to suggest that these are necessarily interesting practical robotics tasks, they are benchmark evaluation tasks for RL algorithms.
>
> _Shooting methods provide exploration that the gradient-driven collocation methods do not allow for._
> We agree that stochastic methods as opposed to pure gradient methods may provide better exploration properties. However, we note that these methods can be used with collocation as well, such as in STOMP. We added a reference to STOMP to related work and a discussion of future work in the conclusion section.
>
> _Can planning methods like CHOMP also be realized in the latent space?_
> Our method (LatCo) has many similarities with CHOMP, which is also a gradient-based collocation method. Our paper describes several of the required pieces for using such methods with deep learning methods: namely, learning a latent space, using approximate second-order methods, and relaxing the dynamics constraint in the initial phase of the optimization. Future work may explore the benefits of different versions of optimization-based planning, such as using Hamiltonian optimization (CHOMP, Ratliff’09), stochastic optimization (STOMP, Kalakrishnan’11), or explicit handling of contact points (CIO, Mordatch’12). We added a discussion of future work to the conclusion.
>
> _How would the results compare to ... a classical motion planner?_
> Classical control techniques can likely solve this task if the labels for a ground truth state descriptor are available, and the dynamics of the system are known. In our work, we focus on deep reinforcement learning, which is applicable even when estimating the true state is hard, or when the dynamics are unknown. In particular, for the pushing task, our model can be used even when the material and contact properties of the object and the robot are unknown, as we can learn these parameters directly from data.
>
> _What is the impact of choosing time horizon T?_
> Horizon T was chosen appropriately for each problem such that the problem is solvable within that horizon (e.g. by extending the horizon until the problem is solved). Horizon that is too short might lead to a poor solution that doesn’t take into account the long-term consequences. Horizon that is too long might lead to solving an unnecessarily computationally expensive problem, although we did not observe degradation in achieved total reward with longer horizons.
>
> _What is stochastic about the dynamics?_
> The underlying MuJoCo simulation for our tasks is deterministic for most purposes. However, we use a probabilistic latent variable model following prior work (Hafner’19), which has shown that it outperforms other deterministic models on visual tasks. This could be attributed to the fact that images only contain imperfect information.
>
> _What is the action space? a-max?_
> The Obstacle task has a 2-dimensional action space corresponding to the target positions. The Metaworld environments have a 4-dimensional action space that controls the end effectors deltas and the paddles. The allowed action range ($a_max$) is 1.0 for all tasks.
>
> _Discussing the broader space of collocation methods_
> We note that we have a brief overview of some prior collocation methods in the related work section. We added a note clarifying that our work is similar to Schulman’14 in that we also use an approximate second-order constrained optimization method with hard constraints.

---

### Official Review · AnonReviewer3 · 2020-11-02
**Review: Model-Based Reinforcement Learning via Latent-Space Collocation**

**Rating:** 4
**Confidence:** 4

**Review:**

#### Summary:
In this paper, the authors propose to replace commonly-used shooting-based methods for action sequence planning in learned latent-space dynamics models by a collocation-based method. They argue that shooting-based methods exhibit problematic behavior especially for sparse-reward and long-horizon tasks, as shooting methods do not allow for planning trajectories which (slightly) violate the learned dynamics. The authors propose a collocation method based on Levenberg-Marquard optimization with a scheduled Lagrange multiplier which outperforms two shooting methods (CEM and gradient-based) on a set of robotic tasks.


#### Pros:
- The paper is clearly written and experiments demonstrated improved performance over CEM and gradient descent optimization of actions.

#### Weaknesses:
- The experiments are limited to sparse-reward tasks, it may be interesting to compare the performance of LatCo and CEM on DeepMind control suite tasks (same as PlaNet), also to see how LatCo performs on dense-reward tasks.
- It is unclear why collocation should find goals better than CEM or gradient descent for sparse rewards. If the reward function network learns this sparse reward, there is no meaningful gradient towards the goal for an optimization based method.  CEM seems to have a better chance to find the goal due to randomization of actions. If not reward shaping has been used, why is the learned reward by the PlaNet network useful for collocation?
- Conclusions claims that the approach would be "removing the need for reward shaping", however the task is simplified by the oracle agent for training data collection which uses reward shaping. The manual interaction is shifted from reward shaping to training data augmentation. Please clarify.

#### Recommendation:
The main concern about the paper is that optimization-based collocation might not be appropriate for the sparse reward case for a method that learns to predict reward for states. Hence experimental results are questionable. The rebuttal should carefully address this issue.
The idea is evaluated in a sufficient range of experiments,  although further experiments on standardized benchmarks (DeepMind control suite) would significantly improve the paper. The points raised in weaknesses above should be addressed.


#### Questions for rebuttal:
- See "weaknesses".
- Why not use gradient descent to update the Lagrange multipliers?
- What is the role of $\epsilon$ in the Lagrangian in algorithm 1 / l5?
- How do the terms in the Lagrangian relate to the residual terms? Especially, why does the quadratic action objective in the Lagrangian relate to the residual $\max(0, |a_t| - a_\mathrm {max})$?
- In 6.3, you write "To provide a fair comparison that isolates the effects of different planning methods, we use the same dynamics model architecture for all agents". Is it only the same architecture, or the same dynamics model (at least for the models trained only on the oracle data)?
- What is the task in Sec. 6.4 to generate the plots in Fig. 5?
- Why do the returns get negative if the reward is sparse and positive ?

#### Further comments:
- Rename $\lambda_t$ in eq. 6 to $\lambda_t^\mathrm{dyn}$,  to match l5 of algorithm 1
- What is the value of $\lambda_t^\mathrm{act}$?
- "For the reward objective, we found it convenient to map the reward to  the negative part of the real line with the softplus operation"  sounds confusing to me, I associate negative numbers with the  negative part of the real line. Maybe phrase it like  "For the reward objective, we form residuals by squashing the negated reward through a softplus function".
- Algorithm 1: $T_\mathrm{rep}$ is not defined
- Algorithm 1 / l13: The ELBO is maximized -> gradient *ascent* (with some learning rate) $\theta := \theta + \alpha \nabla ...$
- \emph{} seems to give underlined instead of italic characters (see the references section), this is probably not intended
- Please plot lagrange multiplier values in Fig 5

#### Post-rebuttal comments
- The paper should further elaborate on the smooth reward predictions and how online learning in the sparse reward setting can be possible with LatCo. It seems the method requires a specific initialization/implementation of the reward predictor, for instance, to overestimate rewards so that the method has to explore the areas where reward is overestimated and pull down the predicted reward. The paper should explain how this was implemented. This kind of exploration would be prone to the curse of dimensionality if the state representation of the environment is high-dimensional. The authors should discuss this limitation thoroughly. This might also explain why the tasks in the experiments are limited to 2-dimensional states.
- I wonder about the discretization of the colors in Fig 8. Higher quantization of color should be provided so gradients of the reward landscape can be assessed.
- The paper still does not detail the update rule for \lambda_act

Overall, the author response has addressed some of my technical concerns, but the main challenges are only addressed partially. The paper is still borderline and might need another thorough round of improvement and resubmission to another venue.

---

> ### Author Response · Authors · 2020-11-19
> **Author Response: provided experiment to explain learned reward; online training experiment**
>
> Thank you for the detailed review and suggestions! We have updated the submission with the suggestions (changes in olive color) and an experiment examining the learned reward predictor, and an online training experiment. We provide detailed replies below. Please let us know if this adequately addresses all of your reservations, or if there are other issues that remain to be addressed.
>
> _“Why is the learned reward by the PlaNet network useful for collocation?”_
> As our experiments demonstrate, LatCo is able to optimize a sparse reward using a gradient-based method. We have inspected this phenomenon as the reviewer requested, and we observe that this is due to smoothing induced by the learned reward function. We have added a qualitative visualization of the reward predictor values in Fig 8. Similar phenomena were observed by prior work learning reward functions, such as Singh’19. We further note that a smooth reward is often easy to define since our method does not require this reward to be shaped.
>
> _The task is simplified by the oracle agent for training data collection_
> We selected the original evaluation protocol to isolate the effect of planning from exploration and model learning. To address the reviewer’s concern, we have now added an experiment that trains the agents according to Algorithm 1 (using MPC and online training), without any offline data. We have updated Tab 1 with these results. On this harder evaluation protocol due to the additional exploration problem, our agent still outperforms shooting-based methods, which are not able to solve the robotic tasks.
>
> _“Compare the performance of LatCo and CEM on DeepMind control suite ... dense-reward tasks.”_
> We agree that it would be interesting to apply LatCo to many scenarios, and we expect it to outperform CEM on hard planning tasks in general. In our work, we focus on sparse reward tasks requiring long-horizon reasoning, which is an important scenario in robotics, and in fact is much harder than corresponding dense-reward tasks. Unfortunately, no standard benchmarks exist for this scenario, but we adapted the popular MetaWorld benchmark by adding visual observations to it. Future work will examine other settings like the DM control suite.
>
> _Why not use gradient descent to update the Lagrange multipliers?_
> We found that gradient descent is too slow and a proportional update ensures fast convergence of the multipliers. This is further explained in the “Constrained optimization” paragraph in Sec 5.
>
> _What is the role of \epsilon in the Lagrangian in algorithm 1 / l5?_
> $\epsilon$ is the target dynamics violation. We found that using a small non-zero $\epsilon$ (1e-5) is beneficial for the optimization and ensures fast convergence, as the exact constraint might be hard to reach.
>
> _Why does the quadratic action objective in the Lagrangian relate to the residual_
> We have updated the action term in the Lagrangian to be consistent with the action residual we used in the experiments. We note that the two versions have the same effect of constraining the actions to be within action space limits.
>
> _“Is it only the same architecture, or the same dynamics model” used for the baselines_
> For the old offline data experiments, we used the same model weights. The new online experiments train several different models (with the same architecture) using the different planners for data collection.
>
> _What is the task in Sec. 6.4 to generate the plots in Fig. 5? Why do the returns get negative?_
> We used the obstacle task for Fig 5. The obstacle task reward is the negative distance to the goal, which explains negative rewards.
>
> _Please plot lagrange multiplier values_
> We have added the lagrange multiplier values to Fig. 7 in the appendix.
>
> _Further comments:_
> Thanks for these very helpful comments! We have incorporated them into a new version of the manuscript.
>
> Singh’19, End-to-End Robotic Reinforcement Learning without Reward Engineering.

---

### Author Response · Authors · 2020-11-25
**Authors: Summary of revisions**

We thank all reviewers for the thorough and useful feedback. The reviewers noted that the proposed approach is “a much better idea than using CEM planning or shooting” and the paper is well-written. The reviewers also raised concerns about whether the method can be used with online data collection, why the method is applicable to sparse reward tasks, and suggested additional discussion of prior work.

To address these concerns, we have performed an additional experiment with online data collection in Tab 1, verifying that our method works well in this more challenging setting and outperforms shooting-based methods. We further produced a visualization of the reward model gradient in Fig. 8, and revised the paper to address further comments. We believe these revisions address the raised concerns and improve the paper quality.

---

### Decision · Program_Chairs · 2021-01-07
**Final Decision**

**Decision:**

Reject

**Comment:**

This work applies collocation, a well known trajectory optimization technique, to the problem of planning in learned visual latent spaces. Evaluations show that collocation-based optimization outperforms shooting via CEM (PlaNet) and  shooting via gradient descent.

Pros:
- I agree with the reviewers that this idea makes sense, and will very likely be built on in future work
- the authors have very actively addressed most comments of all reviewers that engaged in discussion

cons:
- I agree with the reviewers that this is a very simple and straightforward application of collocation methods to the visual latent space domain. Furthermore, the chosen tasks are fairly simplistic, meta-world has a variety of tasks, most of which are more complex than the reaching and pushing task that were chosen for this manuscript.
- Even with all the updates, the evaluation is still very shallow. I agree with the reviewers that obtaining results for both settings: a) visual MPC with pre-trained (or even ground truth) dynamics model b) in the model-based RL setting, for which the model is being learned, is important. While the authors have added some of these experiments, a detailed discussion of how the results change from a) to b) is missing.  Furthermore, when using collocation in this MBRL setting, how should dynamics constraints be enforced (should they even be enforced when the model is still really bad?). How does the comparison between collocation and shooting fare when you use dense/shaped rewards for the sawyer tasks? Many questions come to mind, some of which that have been raised by the reviewers, and my main point is that simple idea + in-depth analysis of some of these questions would have created a stronger contribution.
- Alternatively, real system experiments would have increased the significance of this work.
- I don't see any direct references of gradient-based visual latent-space planning (shooting), but related work on this does exist.

In my opinion, a simple straightforward idea is no reason to reject a paper. However, currently, the reader does not learn when collocation should be considered over other trajectory optimization methods, when attempting to plan in a learned visual latent space. And what some of the main remaining challenges are. Because of this I lean towards recommending reject, and would encourage the authors to deepen their analysis of collocation in visual latent space.